# Social Suffering: Indigenous Peoples’ Experiences of Accessing Mental Health and Substance Use Services

**DOI:** 10.3390/ijerph20043288

**Published:** 2023-02-13

**Authors:** Victoria Smye, Annette J. Browne, Viviane Josewski, Barbara Keith, William Mussell

**Affiliations:** 1Arthur Labatt Family School of Nursing, Western University, 1151 Richmond Street, London, ON N6A 3K7, Canada; 2School of Nursing, University of British Columbia, T201-2211 Wesbrook Mall, Vancouver, BC V6T 2B5, Canada; 3School of Nursing, University of Northern British Columbia, 3333 University Way, Prince George, BC V2N 4Z9, Canada; 4Consultant, Burnaby, BC V5H 4E8, Canada; 5Sal’i’shan Institute, 800 Wellington Ave, Chilliwack, BC V2P 6H7, Canada

**Keywords:** Canada, social suffering, structural violence, relational practice, Indigenous health, equity, mental health, substance use, addiction

## Abstract

In this paper, we present findings from a qualitative study that explored Indigenous people’s experiences of mental health and addictions care in the context of an inner-city area in Western Canada. Using an ethnographic design, a total of 39 clients accessing 5 community-based mental health care agencies were interviewed, including 18 in-depth individual interviews and 4 focus groups. Health care providers also were interviewed (n = 24). Data analysis identified four intersecting themes: normalization of social suffering; re-creation of trauma; the challenge of reconciling constrained lives with harm reduction; and mitigating suffering through relational practice. The results highlight the complexities of experiences of accessing systems of care for Indigenous people marginalized by poverty and other forms of social inequity, and the potential harms that arise from inattention to the intersecting social context(s) of peoples’ lives. Service delivery that aims to address the mental health concerns of Indigenous people must be designed with awareness of, and responsiveness to, the impact of structural violence and social suffering on peoples’ lived realities. A relational policy and policy lens is key to alleviate patterns of social suffering and counter the harms that are unwittingly created when social suffering is normalized.

## 1. Introduction

Access to appropriate systems of care is important to achieving overall improvements in the [mental] health status among Indigenous populations in Canada [1,2,3,4]. Despite this, Indigenous mental health has tended to be glossed over by some mainstream health authorities, and other decision-making bodies, specifically in urban areas [5,6,7]. Inequities in access to culturally safe and effective care remain a pressing concern for those Indigenous peoples who live with mental health and/or substance use issues and, in particular, for people whose lives also are shaped by the intersecting issues of poverty, trauma, and violence, and other social and health inequities [5,6]. In this paper, we use the term substance use or substance use issues to also include ‘addiction’, given the general shift away from using the term addiction because of the stigma associated with its use (including the DSM-5).

The mental health inequities affecting Indigenous peoples in Canada are well documented and directly attributable to the country’s history of colonization, ongoing neo-colonial policies, and racism [1,8,9,10,11]. High proportions of the Indigenous population have been marginalized by systemic racism, discrimination, and intergenerational traumas resulting from residential schooling and other colonial policies and practices [6,8]. Despite these realities, mainstream mental health, substance use, and social welfare services, programs, and policies in Canada have tended to evolve without an awareness of the impact of structural violence and social suffering on people’s lived realities [12,13,14,15,16].

Structural violence increasingly is seen in public and population health as a major determinant of the distribution and outcome of health inequities [12,17,18]. It refers to the existence of unequal power, restricted access to resources, and systematic oppression resulting in the denial of basic needs. Closely aligned with social injustice, the arrangements are *structural* because they are embedded in the political and economic organization (institutional inequality) of our social world and they are *violent* because they cause, influence, and govern individual experience; they cause injury to people, deny human rights, constrain human agency, and/or prevent particular individuals and population groups from having the resources necessary to help them reach their full potential [17,18,19]. Political, economic, and social inequalities limit the personal agency of an individual to live a healthy life or to seek care. Failure to address structural violence is, at least in part, related to its insidious and silent nature [12,17,20]. Structural violence is inextricably linked to social suffering, which is defined as the result of not only the direct exercise of political, economic, and institutional power over people, but also of the way these forms of power shape human experience and social issues [21].

To foster the development of policies and practices that are more responsive to the mental health concerns and service needs of Indigenous peoples, it is important to examine how structural violence manifests institutionally to affect Indigenous mental health and well-being and access to care. This article draws on findings from a Canadian ethnographic study in which we explored Indigenous peoples’ experiences of mental health and substance use services in an urban context. The overarching objective of the study was to improve access to appropriate, effective mental health and substance use services for Indigenous peoples, with a particular focus on those most impacted by social and structural inequities.

## 2. The Political and Historical Context of Indigenous Mental Health Inequities

Current issues of Indigenous access to mental health and substance use services and attendant systems of care must be examined within the context of a colonial history in Canada and the resulting paternalistic and authoritarian practices in health care delivery to Indigenous peoples; the provision of mental health and substance use services has been shaped by over a century of internal colonial practices, policies, and politics [16,22,23,24,25]. The systematic subjugation of Indigenous peoples has its origins in the colonial laws and policies enacted upon Indigenous peoples in 1876 in the *Indian Act*. While this framework was built on the pretext of assisting ‘Indians,’ the underlying intention was to civilize and eliminate ‘Indians’ and despite several amendments, this *Act* is still in existence today. The assimilationist intent of the *Act* was pursued at many levels: Indigenous lands were appropriated and reserves were established; residential schools were instituted with the goal of indoctrinating children into the dominant culture (a collaborative effort between church and state); and cultural spiritual practices were outlawed [10,23,26]. Despite this history, it is important to note that many Indigenous people are living well, e.g., as noted in the ‘First Nations Mental Wellness Continuum’ report, “First Nations have maintained their cultural knowledge in their ways of living (with the land and with each other) and in their language. These foundations have ensured First Nations people have strength, laughter, and resilience” [22]—in fact, cultural continuity has been put forward as a determinant of Indigenous peoples’ health [27]. In addition, statistics related to Indigenous health need to be understood as variable; the impact of colonialism, neo-colonialism, racism, etc., has not been the same for all Indigenous people/communities, rather health/well-being has been shown to be associated with variation in access to education, housing, food security, employment, and health care, and protective features such as connectedness to culture—to name several [4,8,28,29].

Regardless, the high rates of substance use, youth suicide, mental health problems, chronic pain, and interpersonal violence in some Indigenous communities are emblematic of the cumulative experiences of systemic discrimination, racism, economic marginalization, and more than a century of assimilationist policies [11,30,31]. Residential schools have been the most often cited cause of the mental health and substance use issues for Indigenous peoples (but certainly not the only cause) [6,7,10,32]. Not surprisingly, residential school experiences have been linked to post-traumatic disorder (most common) and substance use, major depression, dysthymia, and anxiety disorders, among many other health effects [7,33]. Adverse childhood experiences have been well documented as strongly linked to problematic substance use, chronic pain, and a wide range of other health effects in adulthood, including early mortality [34,35].

The overall impact of residential schooling in the form of intergenerational and historic trauma has been devastating [3,5,19,20,21,22,23,24,25]. Research continues to demonstrate major discrepancies in mental health status among Indigenous populations compared with other Canadians [20,27,28]. For example, although there have not been any suicides reported in 60% of First Nations communities between 2011 and 2016, suicide rates remain significantly higher for First Nations, Métis, and Inuit than for non-Indigenous Canadians; the suicide rates and disparities were highest in youth and young adults (15 to 24 years) among First Nations males and Inuit males and females and are particularly alarming given that nearly 40% of Canada’s First Nations people are under 25 years of age [11]. Notably, in a study by Chandler and Lalonde [36] in 1998, rates of suicide were found to vary dramatically across communities and were related to markers of cultural continuity, i.e., what they deemed as ‘protective’ factors such as land claim negotiations, self-government, education services, police and fire services, health services, and cultural facilities.

Higher rates of illicit drug and other substance use also have been reported [37,38]. Furthermore, although a smaller proportion of Indigenous peoples consume alcohol than in the general population, the rate of problem drinking is higher in the Indigenous population [39]. Health and social disparities parallel mental health inequities. First Nations, Inuit, and Métis are also much more likely to die younger than non-Indigenous populations [40] and live in inadequate housing [41] or to be homeless [42,43,44]. According to Thistle, Indigenous homelessness is “a human condition that describes First Nations, Métis and Inuit individuals, families or communities lacking stable, permanent, appropriate housing, or the immediate prospect, means or ability to acquire such housing” [45] (p. 6). There is a gross overrepresentation of Indigenous homelessness in urban settings in Canada [43]; studies indicate that Indigenous people are 8 times more likely to be homeless than non-Indigenous people and represent between 10% and 80% of the total homeless population in large urban centres [43,44,46]. People with severe substance use issues and/or mental illness also can be found in this group—they make up anywhere from 33% to over 60% of the overall homeless population [47]. Poverty intersects with mental illness and substance use to render many vulnerable to homelessness; poverty associated with lower educational attainment and less adequate training for many Indigenous peoples makes living in urban settings more difficult and susceptibility to homelessness more likely [42]. In addition, Indigenous women and children/youth experience disproportionate rates of violence and abuse compared to Indigenous men and non-Indigenous people, children, and youth [30,42]. The greater levels of violence experienced by Indigenous women can be seen as a consequence of the intersecting effects of colonization, racism, classism, and sexism, and both reflect and contribute to the greater risk of homelessness lived by Indigenous people [15,30,48,49].

Social suffering is both socially produced and mediated by historical, political, economic, and cultural factors, i.e., structural violence [17,19,21]. Mental health institutions and policies support, often unwittingly, the embeddedness of an assimilationist ethos, putting Indigenous people at risk of not having their mental health care needs recognized and met [16,50,51]. Indigenous perspectives and issues continue to be largely excluded vis à vis the intersection of mental health programs and services designed in keeping with dominant cultural (biomedical) views of mental health and illness, the ongoing jurisdictional debate regarding *who* is responsible for Indigenous mental health, especially within urban geographies, and an increasingly neo-liberal health care climate [16]. This is the political and historical context within which we aimed to explore Indigenous peoples’ experiences of mental health and substance use services in our study. As we discuss in this paper, Indigenous peoples’ experiences of mental health and health care cannot be divorced from the social contexts of their lives. Service delivery that aims to address the mental health concerns of Indigenous peoples must be designed with awareness of, and responsiveness to, the impact of structural violence and social suffering on Indigenous people’s lived realities, including the intersecting effects of past and ongoing colonialism, racism, and other forms of discrimination.

## 3. Methodology and Methods

### 3.1. Overview of the Study

The findings discussed here were derived from a critical ethnographic study informed by Indigenous epistemologies and, in particular, the notions of relationality and self-determination. For example, all aspects of the research, including the design, were guided by the investigative team in consultation with an Indigenous Community Advisory Committee, which included community leaders, patient representatives, and clinical experts in Indigenous mental health, and central to the study was ongoing consultation with an Elder and Indigenous leaders. The investigative team, comprised of Indigenous and non-Indigenous academic researchers, community leaders, including an Elder, and community-based mental health agencies, was led by a non-Indigenous researcher with long-standing, ongoing relationships in the inner-city community where the study was located.

The research was guided by ethical guidelines developed for research with Indigenous peoples [52,53], and formal ethical approval was obtained from the Research Ethics Board of the university that served as the primary affiliation of the principal investigator on the study. The study was designed to (1) explore Indigenous peoples’ experiences of mental health and substance use services; (2) analyze those experiences within wider institutional and sociopolitical contexts to understand how these contexts shape the provision of mental health and substance use services; and (3) use the research findings to provide policy and practice directions for how to improve access to services that are safe and more responsive to the intersecting mental health needs of Indigenous peoples.

Postcolonial theories served as an interpretive lens for understanding how racism, colonialism, classism, gender, and historical positioning intersect to produce and sustain inequitable social relations within and beyond health care [54,55,56]. The prefix ‘post’ in postcolonial theorizing is not intended to imply that colonialism is over but to direct attention to the legacy and ongoing manifestations of colonialism, and how this continues to shape people’s lives and life opportunities, including health [55,57]. Postcolonial perspectives are thus particularly relevant to health research with Indigenous peoples [54,56] and are drawn on by Indigenous and non-Indigenous scholars to re-centre Indigenous voices and perspectives within social inquiries aimed at addressing the past and ongoing effects of colonial and other forms of oppression [16,58,59].

### 3.2. Participants and Data Collection

Purposeful sampling was used to recruit persons with lived experience (PLE) of accessing mental health and/or substance use services and providers from five community-based mental health and primary health care agencies. These sites were chosen because they were known to serve high proportions of Indigenous peoples. Data collection involved (a) in-depth individual interviews with 18 Indigenous persons (16 First Nations; 3 Métis) accessing mental health and/or substance use services at 1 of 3 sites (n = 10-7-1); (b) 3 focus group interviews involving an additional 21 Indigenous and non-Indigenous persons accessing services at 1 of 3 sites (n = 9-6-6) to assist in understanding the unique experiences of Indigenous participants; (c) in-depth interviews with 22 health care providers working at 1 of 5 health care settings (n = 8-2-6-5-1) (including outreach mental health workers, counselors, psychologists, psychiatrists, nurses, and social workers) to provide another perspective on services and service delivery as part of triangulation processes; and (d) participant observations within the health care settings.

The PLE who participated in this study (n = 39) included 19 men and 20 women (age range 22 to 59 years). At the time of the interview, all PLE were living on low incomes, and the majority (79%) was unemployed and existing on meager welfare or disability payments. Seventy-two percent of the PLE participants were living in unstable housing, including single room occupancy hotels, social housing, and shelters. Presenting issues for these participants included co-occurring illnesses such as schizoaffective disorder, mood disorders, depression, anxiety, suicidal ideation, alcohol and drug use, HIV, Hepatitis C, and post-traumatic stress disorder associated with complex trauma. Several participants were residential school survivors and most had a long history of trauma, beginning in early childhood and, for many, continuing into the present. 

Recruitment was facilitated through on-site liaisons, who provided information regarding the study, including contact information, to potential participants. Participants contacted the researcher who then discussed the study and obtained consent. Gift cards were provided as honoraria to all PLE. Interviews with the PLE were conducted by the academic researchers in a safe setting chosen by the participants and typically lasted one to two hours; research interviewers explored their experiences receiving care, including the reasons they sought care in the particular setting, their assumptions and expectations about that care, their experiences of seeking care elsewhere, and their interest in Indigenous healing practices. In-depth interviews with health care providers focused on their understanding of why and how Indigenous peoples seek care in their setting. All interviews were audiotaped and transcribed verbatim for analysis.

### 3.3. Data Analysis

Using procedures for qualitatively derived data, an interpretive thematic analysis was conducted [48]. Interview transcripts were repeatedly read to identify recurring, converging, and contradictory patterns of interaction in the data, including key concepts and possible linkages to theoretical perspectives. Data were organized and coded using NVivo, a qualitative research software tool [49]. As more data were collected and analyzed, coding categories were refined and taken to a higher conceptual level, shifting analysis to a more abstract representation of the ideas and themes expressed through the data. The investigative team members met regularly to discuss the data and analysis; ongoing dialogue within the team and then with a subset of participants assisted with the generation of primary themes and broader analytical insights.

Credibility, as a criterion for rigor in qualitative analysis, was continually evaluated by discussing the emerging analysis with members of our investigative team and the Indigenous Community Advisory Committee. These stakeholders concurred that the themes reflected in the data resonated with their experiences and their interpretations of those experiences. The trustworthiness of the analysis was also optimized through the triangulation of Indigenous and non-Indigenous client participants, staff, and observational data [60].

## 4. Results

The findings of this study underscore the importance of understanding structural violence as a key determinant of health. While many of the experiences of structural violence depicted in the following excerpts are not necessarily specific to Indigenous peoples, our analysis illustrates that for Indigenous peoples, structural violence is often an ongoing reality and experienced in multiple complex and intersecting ways. The results are organized around four intersecting themes: normalization of social suffering; re-creation of trauma; the challenge of reconciling harm reduction with constrained lives; and mitigating suffering through relational practice.

***Normalization of social suffering***. One overriding theme in the interviews reflected the extent to which social suffering was enmeshed in participants’ experiences of everyday life, mental health and well-being, and accessing care. Consistent with the lives of many marginalized people residing in core urban areas in Canada, most of the participants in this study were living in extreme poverty, homeless, or living in inadequate, unsafe, or unstable housing, and were experiencing the concurrent patterns of ill health that plague people living in poverty. When inadequate social conditions intersect with people’s emotional or mental health problems, the potential for exacerbating physical and emotional angst is extreme. The following interview excerpt with an Indigenous woman illustrates this.


*… I’ve lived there for a year and it’s starting to really get into my system because of the mold and the pesticides in the walls…you can poke a hole and all this dust will come out…so I want to get out of that building quick because I know it’s in my lungs now because I wake up in the morning and it’s like…all gurgling in my chest, I’m like, oh no, this isn’t good, you know, I’ve got to get out of there.*


The woman in the above example lived with HIV illness, Hepatitis C, mental illness, and substance use issues. Her everyday existence was mediated by the ability or inability to access housing/shelter, food, and treatment. Like this woman, most people in this study lived with serious co-morbid illness rendering them vulnerable to other illnesses. Life was further marked by isolation, an experience of many in this study, as articulated by an Indigenous woman in the following: “When I get sick, nobody comes to see me; I literally sit there and starve for a couple of days.”

Unstable and unsafe housing, homelessness, and isolation created many challenges for the participants in this study. A nurse, who worked in a mental health clinic, noted that the first discharge option from a provincial mental facility for an Indigenous patient she was following was a place well known for being a slum hotel. She said, “The [name of hotel] is one of the more despicable places that has rats running through the hallways… it’s rodent infested, bug infested and very, very dangerous.” Like many Canadian urban areas, despite recent efforts to procure more housing, the lack of safe, affordable housing remains at crisis proportions; this creates a particular set of vulnerabilities for people living with mental illness or addictions [61].

Housing options often are constrained by the client profile, i.e., people living with manifestations of profound illness are often excluded from healthy living environments. The nurse above relayed a story of another mental health client who had been in an extremely run-down hotel. Apparently, he would write on the walls with felt pen (he did not talk), leaving him with few places where he could meet the housing criteria. In the following interview excerpt, she notes his response to a new apartment, which was a much higher standard of housing than he had lived in previously.


*…He always writes on the walls and so [name of housing official] said, okay, well, maybe if you just use pencil instead of felt pen what do you think of that, …okay, so it’s always options, it’s never you have to do this, you have to do that and he wrote one word on the wall [when he first arrived in his new place] and it was ‘comfort’… he uses far less drugs and he eats regularly now.*


In another example, an Indigenous female participant speaks to her experience of waiting in her physician’s office.


*One of the things I do have problems with is she [doctor] is too busy. I went there my last appointment, my appointment was at one, I didn’t get out of there till five and I was really, really sick too. I had a high fever, I think my temperature was 39 or something, I can’t remember but I didn’t care, I just kind of laid back on her couch and stuffed my face with what they had sitting there…I drank a whole pile of juice.*


Many of the focus group participants in the study raised the issue of waiting, which has been normalized by several other participants in this study. The woman above had a trauma history associated with the legacy of colonialism (residential schools) and co-occurring illness, was HIV+, and happened to be very ill. Within this social context, it is not surprising that she normalized the experience of waiting in much the same way as stigma and discrimination have been normalized for many in this population. The issue of waiting to be seen also draws attention to the lack of places that people marginalized by inequity feel they can access to obtain care, given their issues, and in some cases, their appearance as poor people or as people who are living on or near the streets. These findings resonate with other research conducted in urban settings in Canada, where Indigenous patients who are visibly poor find it challenging to seek help in places without encountering judgment and stigmatization [62,63]. Thus, the need to wait, and the acceptance or normalization of waiting, takes on additional significance when people perceive there are limited alternate places to access help and/or they have normalized being treated poorly and come to expect it. If we conceptualize health care as a form of a social relationship rather than simply a service, where we take into consideration the power dynamics and the social, political, and historical origins of inequity to health care access for Indigenous peoples and its health impact, waiting is not an innocuous event, regardless of the underlying justification for the wait. Access is not simply the availability of services and providers but the delivery of services at the point of care—it is a relational moment.

***Re-creation of trauma*.** Most of the participants shared stories about the way everyday health care practices created discomfort or distress. These experiences, in many ways, represented an extension of the marginalizing experiences that many participants described in everyday interactions outside of health care. In the following interview excerpt, an Indigenous female participant recounts a recent hospital experience.


*I shared a room with a guy… and then down the hall my cousin is in the fucking hospital, she’s got this big, giant room to herself, and then on the other side of the corridor, a friend of mine from [name of location] is in the hospital too and she’s sharing that room with a guy, like why didn’t they put us together?*


This participant did not feel she had the ability to have her room changed—she was an Indigenous woman who was in the hospital because of an abscess associated with intravenous drug use, was living in abject poverty, was HIV+, and had a long history of abuse which began in childhood. Her social location made her cautious about what she said. As she explained later, she feared she might be forced to leave the hospital if she protested too vigorously.

Many health regions in Canada have endorsed the practice of co-ed rooms in hospital settings. This decision fails to take into account the multiple forms of past and/or current trauma and violence that shape the lives of many Indigenous women [15], including those participating in this study. For the woman whose voice is being represented in the above excerpt, this hospital visit was a re-creation of the trauma she endured most of her life, beginning with her experience in residential school. Hospital policy intersects with the traumas that people continue to live with. Here, policy decisions can intersect with lived experiences to exacerbate trauma and distress for those who are most vulnerable. When we understand interpersonal and structural violence as intersecting forces and the impact of violence as ongoing and cumulative, the risk of re-traumatization can be seen more clearly.

In another example, and in a slightly different vein, an Indigenous man provides his perspective on the health care system.


*Within the system there is some prejudice people in there and I try not to get too mad with them when I find out that they’re prejudice, they don’t like Natives and they don’t like drug addicts.*


In describing his health care experience, this participant provides insight into the powerful intersecting stigmas that contribute to particular inclusionary and exclusionary experiences. His perception of the provider response to him is mediated by his experiences as a First Nations man living with substance use issues. In another example, in response to the researcher’s question regarding why he did not access mental health services given that he was living with significant concerns regarding his mental health, another Indigenous male participant noted, “I’ve already got HIV, now I’m crazy too?” As this man further explained, being Indigenous and living with HIV and the associated stigma and discrimination he had encountered in health care was difficult enough. He did not feel he could endure the stigma he perceived he would encounter if he accessed mental health services, regardless of the symptoms of mental illness he was experiencing at the time. Many of the people in this study expressed this concern; often perceived to be the most stigmatizing condition was living with a mental illness.

This interview excerpt illustrates how perceived stigma and discrimination mediate access to the health care system—an issue for many people living with mental illness and substance use, regardless of their ethnocultural background. This participant points to the way in which (dis)ability and illness are both understood as marginalizing and render access difficult. However, stigma and discrimination associated with mental illness (and substance use) then intersect with racializing practices and poverty to become a powerful oppressive force for many people. Many of the participants in this study connected their negative experiences to how they were ‘read’ by providers—as poor, mentally ill, addicts, “Native,” HIV-positive, and so on.

As noted earlier, most of the patients in this study had been profoundly affected by trauma in their lives, and many in the study continue to live with violence. This reality renders people particularly vulnerable to negative experiences and encounters in the health care system. These experiences often created discomfort, fear, shame, and a sense of alienation from those services.

***The challenge of reconciling constrained lives with harm reduction*.** Most participants had co-occurring illness, mental illness, and substance use, and difficulty in accessing treatment for one or both issues. Being free of substance use is a prerequisite for many treatment and housing programs, yet detoxification (detox) was one of the most difficult services to access in a timely fashion. An Indigenous male participant in a focus group forum discussed this in the following excerpt.


*An addict can’t wait because you can sit there, you can’t wait, when you make the decision to go in and actually detox yourself and clean up, [and] if you’re not accepted right away… it doesn’t take long to get discouraged because you’re inundated with drugs down here…it doesn’t take long before you forget about all those great goals you had…*


The so-called *detox paradox* is a well-known issue within the core area in which the study occurred. It is further complicated by the geographic location of detox programs and neighborhood boundaries. In the following focus group excerpt, another Indigenous male participant describes this problem.


*Well, it’s just the people down here, you know everything in this area is accessible within a two-block radius. You leave out of here; you get an apartment up on [name of location] and you’ve got to walk twenty blocks because the bus driver is not going to give you a free ride. Everything is accessible down here; they made it that way so we’ll stay in this area.*


Here, waiting for detox services intersects with where, how, and to whom services are offered. For example, several of the participants in this focus group noted the problem of living in what they referred to as “The Box”—a section of the city that is approximately a four-block radius where most of the services for this population exist—a concern also noted by several health care providers working there. As one health care provider noted, the geographic location of services in this area not only confines clients but makes it difficult for them to shift their lives. In a similar vein, several providers noted that clients who were no longer using substances and particularly vulnerable to using drugs faced ongoing harassment by persons selling drugs as they accessed services in the area. As a substitution/maintenance therapy for opioid addiction, within Canada, the Canadian Research Initiative in Substance Misuse developed the National Guidelines for the Clinical Management of Opioid Use Disorder which recommend buprenorphine/naloxone (bup/nal) as a first-line medication for most individuals in Canada and methadone as a second-line treatment if bup/nal is not appropriate or if treatment has limitations or is ineffective [64]. However, at the time of this study, methadone was the most common treatment option, and for those persons in this study, the only treatment option offered. Methadone maintenance clients not only faced life limitations associated with their geographic location, but they also faced limitations associated with the protocols associated with methadone administration. A focus group participant shares this in the following interview excerpt.


*Yeah and it’s terrible because I can’t go, like I can’t go camping because they won’t give me [enough Methadone] to go camping, you see that’s what wrecks it all right?... it’s worse than jail.*


The policies related to MMT limited life choices for several of the participants in this study. In addition, participants repeatedly questioned if inability to obtain help was based on their identity as an Indigenous person. An Indigenous female participant raises this issue in the following excerpt.


*I thought they were kind of, I don’t know, after I’d been a couple of times into the hospital and what they would do, they’d keep me in there for maybe two or three days and then just let me back out. I was homeless and they would let me back out on the street with no help and I thought that wasn’t very good…I don’t know if it’s because I was Native they didn’t really want to help me, that’s how it felt.*


Experiences of discrimination and stigma were complex and multi-layered; however, they were particularly salient for participants who lived with chronic pain. In the following interview excerpt, an Indigenous male participant describes his experience with pain management.


*Everything he’s talking about is true about the doctors down here, the lack of care for a person in pain; because we’re drug addicted they have it in their mind that we’re not giving you drugs of any kind, regardless of what kind of pain you’re in… There’s certainly is a discrimination, not a small one, a big one.*


***Mitigating suffering through relational practice***. In response to the question regarding what, if anything, people found helpful in the system of care, most of the participants spoke about the support of particular health care providers and aspects of the agency where they were accessing health services. For several participants, this was as simple as being greeted with a smile at the reception desk or experiencing the surprise of being offered a cup of coffee (in one agency, the receptionist got the coffee for the client); however, for most of the participants, positive elements of care were related to practices where the provider behaved, as one participant noted, “outside the box,” that is, they behaved in ways outside of normal practice, such as reflected in the following interview excerpt.


*Like myself I’ve been missing my appointments for umpteen times now so now the pharmacist got involved and they’re going to drive me, they’re going to pick me up and take me to my appointments there.*


According to health care providers, missed appointments were common for many of the clients using their services in a couple of the access sites—substance use, mental illness, HIV/AIDS, Hepatitis C, diabetes, and other co-morbidities intersected with poverty to disrupt the lives of many of the people who accessed services. In the case of MMT, the particular pharmacist, as referred to above, often sent someone out to get people who may be having difficulty getting to the service, a form of assistance greatly appreciated by several participants. He is also known to provide food for people, such as those who need to take food with their medications, and often celebrates birthdays with his clients, e.g., he usually has cake in his fridge [65]. According to one of the local physicians, he “does not enable people, he supports” them—she notes the high level of “professionalism” and “respect” evident in this pharmacy [65]. Social support is a well-known determinant of health and one valued by this pharmacist and many of the providers in an adjacent agency. This is also reflected in the following interview excerpt with another health care provider.


*… and access to primary health care they have, it’s like a trust… it’s a place that they can come that they’re accepted and welcomed and, ‘hi Joe’ and ‘how are you’ and they’re treated like anyone else and just the fact that they have their own doctor, they know their nurses, everybody knows their name, people know their histories, know their stories and talk to them and it feels good.*


Several of the participants who accessed this research site spoke about the importance of integrated care and feeling comfortable in that setting. Relational approaches to practice are understood and valued by several of the providers. There was a recognition of social context as a highly powerful force which shapes both health and health care reflected in the care provided. An Indigenous client participant speaks to the impact of the care in this setting on his life in the following excerpt.


*The referral service here is great, I mean like they refer you to other agencies, housing, Native housing… When I first started here I started on my medications and they referred me to [name of program] where I went for my HIV medications so, yeah, they’re really good at referring, referring us to other agencies…It’s really comfortable coming here…*


Similarly, an Indigenous female participant notes the importance of connection with her doctor and the recognition of social context in the following excerpt. “She lets me know that [if she cares] by asking about me when I’m not around, stuff like that, she asks my outreach worker… I like when she asks me about my traditional practices—they are very important to me.”

In the agency being accessed above, participants noted the importance of relational approaches, i.e., although the interpersonal aspect of the client–practitioner relationship was important, it was the actions that reflected an understanding of the contextual features of the clients’ lives that made a difference to the participants; these were not simply acts of respect, kindness, care, etc., rather they also were acts that reflected knowledge of colonial policies and practices and historical/intergenerational trauma and that people’s lives were often very difficult in which poverty, hunger, and other social issues were an everyday reality for most of the clients accessing services across these sites. In addition, in the agency described above, several of the clients had worked within the setting for wages and case workers were actively seeking housing, education, and other social support with their clients.

## 5. Discussion 

The study’s findings highlight the complexity of accessing mental health and substance use services for Indigenous peoples whose lives are shaped by structural inequities and structural violence. For study participants, intergenerational trauma, violence, homelessness, isolation, racialization, stigma, discrimination, and poverty were all significant determinants of health. Yet, one of the most striking findings was the extent of, lack of attention to, and understanding of social suffering in health care environments. This has resulted in the normalization of social suffering. In turn, trauma can sometimes be re-created within systems of care, particularly when approaches intended to minimize harm do not fit for people who are most marginalized by poverty and other forms of social inequity.

For example, the data discussed in this paper underscore the need for adequate and safe housing as one of the cornerstones of effective care for people whose lives are shaped by the intersections among poverty, trauma, mental health, and/or substance use issues [42,43,44,46,66]. However, within the mental health care environment, the root causes of social suffering tend to be largely unrecognized, and thus, social suffering is poorly understood. This is particularly true for the complex intersecting dynamics that shape Indigenous peoples’ experiences of homelessness, mental health issues, and problematic substance use [42,45,66]; urban Indigenous populations are disproportionately represented in the low income group and experience significantly higher rates of poverty and homelessness, in particular street homelessness [42,44,46,47]. 

The legacy of colonial health care continues to be evident in various ways, particularly through the social suffering inflicted by contemporary health and social policies and practices (e.g., current discharge practices, policies for co-ed rooms by hospitals). Research demonstrates that Indigenous peoples face serious access problems stemming from discrimination, stigmatization, and disadvantage based on race, gender, and class [3,4,50,62,63]. These sociopolitical realities, manifested structurally within institutions, shape Indigenous peoples’ everyday social experiences and access to routine health care services. Attention to these social context(s) in which people’s experiences of accessing mental health and substance use care emerge is important to identifying the root causes of social suffering, and the impact of structural violence [2,67]. In particular, our research shows that when social suffering, such as inhumane housing conditions, is normalized, the social realities of patients’ constrained lives are often rendered invisible in systems of care, resulting in negative health care experiences, unmet health and social needs, and sometimes the re-creation of trauma within systems of care.

Trauma may be unwittingly re-created when providers are not attuned to the multiple intersecting forms of social inequity that impact health and well-being [2,15], and stigma and discrimination within systems of care remain unchecked. For many participants (Indigenous and non-Indigenous), perceived stigma and discrimination associated with homelessness, drug use, mental illness, and other diagnoses was a common experience in their interactions with health care providers and mediated people’s health care experiences, enactment of agency, and in/ability to access mental health and substance use services and other key supports. Yet, for Indigenous participants, experiences of layered stigma and discrimination were also mediated by their cultural and racialized identities and the multiple negative ways in which Indigenous peoples have been read and treated historically, so that participants repeatedly questioned if inability to obtain help was based on their identity as an Indigenous person. Although not all of these perceptions of stigma may have been cases of enacted stigma, the impact of perceived and/or enacted stigma for people who are affected by multiple social and structural inequities can be profound as multiple stigma intersect [15,68,69].

As noted earlier, several study participants cited societal and institutional stigma associated with drug use and racialized constructions of their identity as an Indigenous person to explain the discriminatory treatment they had encountered when trying to access pain medication. In this way, our findings support other research, which shows that stereotyping, bias, socioeconomic considerations, and social factors (e.g., age, gender, class, ability, substance use, etc.) intersect to create inequities in pain management [3,67,70,71]. There is ample evidence that people who are considered a member of an ethnic/racial minority population are significantly less likely to receive prescriptions for controlled substances, such as opioids, for treatment of pain, particularly chronic pain [70,71].

While the reluctance to prescribe opioids to people diagnosed with problematic substance use may be warranted in individual cases, unexamined biases in decision-making processes in clinical interactions carry with them high personal and societal costs that are disproportionately burdensome for those whose lives are constrained by multiple disadvantages [71]. Studies examining adverse childhood experiences (ACES) have long shown significant health effects of childhood and cumulative trauma, e.g., childhood trauma either in the form of physical, sexual, mental, or psychological abuse is strongly linked to chronic pain and problematic substance use [34,35,71]. At the individual level, undertreatment of pain causes unnecessary physiological suffering with potentially significant negative outcomes for long-term recovery, mental health, and quality of life; inability to access effective pain management may engender mental, physical, and spiritual suffering when patients perceive their treatment to be a function of discrimination and racism, and further undermine individual health and well-being and trust with the health care system [67,70,71].

Findings from this and other research show that patients from racialized groups who experience stigma, discrimination, and marginalization in their wider social experiences are particularly attuned to the attitudes and practices they encounter in health care [2,3]. For people most marginalized by social and structural factors, and who live with both substance use and chronic pain, this may mean that street drugs and alcohol are often the only accessible resource for pain management. Responding solely to chronic pain or substance use as separate entities, rather than the intersecting health effects of both, can compromise overall care and prognosis of patients [72]. Thus, research supports the integration of services to address co-morbidities in substance use programs [67,71,72]. At the societal level, discriminatory decision-making practices, such as those observed in pain management, and its attendant consequences for health and well-being exemplify how stigmatization, discrimination, and disadvantage across multiple axes of differentiation manifest structurally at institutional and societal levels as social determinants of health and health inequities [67,69,73,74].

Our analysis demonstrates that policies and strategies are needed to respond to people marginalized by social and structural inequality in ways that counter the harms that can arise from inattention to social suffering. When clients experience trauma within systems of care in the form of racism, discrimination, and so on, social suffering is potentially intensified [2,15]. This has particular salience for Indigenous peoples in Canada, many of whom experience barriers to accessing mental health care in urban settings [2,15,50].

Among those participants who did have positive experiences, what was so highly valued and meaningful were relational approaches to practice. Although interpersonal relationships were notably important to the participants, practice that was most significant to participants reflected an understanding of what was significant to people in the context of their everyday lives, such as the pharmacist’s practice of seeking out people who did not make it in to receive their medications (e.g., methadone) and providing food or the mental health nurse’s strategy to ensure improved housing for the man who wrote on his walls as a means of communicating with others. These practices reflect an understanding of the everyday challenges people face when accessing services and how people’s individual capacities and agencies could best be supported to open up new opportunities, possibilities, and choices. For many participants in this study, practices such as these assisted in the formation of positive alliances with providers and improved access to resources to support health and well-being.

Overall, our findings underscore the importance of approaches to policy and practice that are equity-oriented and address both individual and structural relations. Building on previous research on equity-oriented care [2,75,76], three intersecting key dimensions of equity-oriented health care identified and tested include (a) trauma- and violence-informed care; (b) culturally safe care; and (c) harm reduction [67]. Alongside the notion of ‘harm reduction,’ our most recent work has included the notion of ‘substance use health’ which acknowledges that substance use is a regular part of most adults’ everyday lives and practices and that substance use can be health promoting for some; however, ‘substance use health’ does not promote substance use, rather it shifts our attention away from substance use as a moral issue to substance use as a health issue [77,78,79,80]. Additionally, our work demonstrates that health equity interventions can be implemented at multiple levels and are associated with improvements in key health outcomes, including better life quality, less chronic pain, and fewer depressive and post-traumatic stress symptoms over time [2,67].

In addition, Downey has put forward the concept of an IND-equity [28]. Situated within a self-determining, rights-based perspective, this model draws attention to the glaring health inequities associated with colonialism, neo-colonialism, and racism and the need for addressing the unique features associated with the social, political, and historical determinants of health (i.e., structural violence) experienced by First Nations, Inuit, and Métis in a culturally safe way [28]. Recent events in Canada surrounding the tragic death of an Indigenous woman in a Quebec hospital, Ms. Joyce Echaquan; the investigation into emergency department staff playing ‘games’ to guess blood-alcohol levels of Indigenous patients in British Columbia; and the over a decade-long investigation into the death of an Indigenous man in Manitoba, Mr. Brian Sinclair, have brought renewed attention to the harms of Indigenous-specific racism—an everyday reality in Canadian society [81,82,83].

Fundamentally, equity-oriented health care is about creating culturally safe and respectful care environments and processes while tailoring care to fit with the needs, histories, and contexts of the populations served. The construction of safer and more respectful care builds upon the concept of trauma-informed care (TIC) and the creation of the concept of trauma- and violence-informed care (TVIC) that explicitly foregrounds violence (the ‘v’) and addresses the intersection and cumulative and ongoing effects of interpersonal (e.g., child abuse, intimate partner violence) and structural (e.g., poverty, stigma) forms of violence on peoples’ lives and health [2,67]. It requires that health care providers adopt practices consistent with key elements of equity-oriented health care, and that organizations develop systems and policies that enable this work [67,73]. For example, “sharing a room with a guy” in a hospital is not an innocuous event for many people, and for some, it carries more risk than for others. Policy decision makers need to consider the consequences of their decisions for people who may be particularly vulnerable to negative outcomes, and health care providers need to advocate for practices in their organizations that actively minimize the risk of re-creating trauma in the everyday provision of services [2,67,71,75].

## 6. Conclusions

In this paper, we conclude by arguing for the development of critical and political consciousness in health care environments to challenge sustaining ideologies, institutional discourses, and predominating practices that perpetuate the status quo in mental health and substance use services in Canada and other nations. Locating Indigenous mental health and mental health care within its wider historical, social, political, and economic context can assist policy decision makers and practitioners to contribute to equity more fully in the area of Indigenous mental health.

## Data Availability

Not applicable.

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
