# Peer review of "Social Suffering: Indigenous Peoples’ Experiences of Accessing Mental Health and Substance Use Services"

_ijerph, 2023, doi:10.3390/ijerph20043288_

Round 1

Reviewer 1 Report

I found this manuscript to be well-written and organized. Very important to understand the underlying trauma and the social suffering experienced by Indigenous people with substance issues. This is often overlooked and can lead to many misdiagnoses. Thank you for focusing on this important issue. 

Author Response

Thank you for the positive feedback. A minor spell check has been completed.

Reviewer 2 Report

Dear authors,

The topic is challengable and suche research is necessiry to sensibilize the vulnerable populations.

My tiny suggestion - to those direct quote (without " ") - wil be better perceived if they are in italic.

Another point is a methodological issue - you mention NVivo, but I have not seen any analyses, results perfomed with it.

You report as "narrative analyses" with quuotes, which is Ok, but the general analyses of all participants is missing - categotries met and their frequencies.

Or if you leve as it is, may be should be specified more clearlier they use of NVivo programme.

With best regards,

Reviewer 3 Report

The article is very interesting and addresses a very relevant issue in the field of indigenous mental health in Canada from a postcolonial and intersectional approach.

The literature review is very well documented, current and relevant.

In terms of methodological aspects, the sample of participants is noteworthy, as it has different types of participants and voices, both indigenous and health care providers, which allows for greater validity of the study and for the triangulation of data. This makes the qualitative study very rich and provides very interesting data on the experiences of the indigenous people and health care workers interviewed. The use of individual interviews and focus groups of five communities also stands out. In the section on participants, it might be useful to include a table describing some characteristics of the population studied (e.g. age, gender, type of addiction/problem, current situation, type of housing, current treatment received, etc.). However, this is left to the researchers to decide.

The results show the complexities of facing structural injustices, marginality, social inequity and systemic violence for indigenous people in the systems of care. The four intersectional themes described have a lot of interest not only for researchers but for mental heath providers and policy makers. 

The conclusions and implications in terms of social policy and the importance of promoting greater social awareness of these issues are very interesting. 
